# SIRT1 Signaling Is Involved in the Vascular Improvement Induced by Moringa Oleifera Seeds during Aging

**DOI:** 10.3390/ph16050761

**Published:** 2023-05-18

**Authors:** Valeria Conti, Joseph Iharinjaka Randriamboavonjy, Herintsoa Rafatro, Valentina Manzo, Jessica Dal Col, Amelia Filippelli, Graziamaria Corbi, Angela Tesse

**Affiliations:** 1Department of Medicine Surgery and Dentistry, Scuola Medica Salernitana, University of Salerno, S. Allende, 84081 Baronissi, SA, Italy; vconti@unisa.it (V.C.); jdalcol@unisa.it (J.D.C.); afilippelli@unisa.it (A.F.); 2Clinical Pharmacology Unit, San Giovanni di Dio e Ruggi d’Aragona University Hospital, San Leonardo 1, 84131 Salerno, SA, Italy; vmanzo@unisa.it; 3INSERM, Université de Nantes, CHU Nantes, CNRS, L’Institut du Thorax, CEDEX 01, F-44000 Nantes, France; randria_njaka@yahoo.fr; 4Laboratoire d’Évaluation Pharmaco Clinique (LEPC), Institut Malgache de Recherches Appliquées (IMRA) Fondation Albert et Suzanne Rakoto-Ratsimamanga (FASRR), Avarabohitra Itaosy, Antananarivo 102, Madagascar; prof.rafatro@gmail.com; 5Department of Translational Medical Sciences, University of Naples Federico II, 80131 Naples, NA, Italy

**Keywords:** vascular aging, endothelial dysfunction, nitric oxide, phytotherapeutics

## Abstract

Vascular aging is linked to reduce NO bioavailability, endothelial dysfunction, oxidative stress, and inflammation. We previously showed that a 4-week treatment of middle-aged Wistar rats (MAWRs, 46 weeks old) with *Moringa oleifera* seed powder (MOI, 750 mg/kg/day) improved vascular function. Here, we investigated the involvement of SIRT1 in MOI-induced vascular improvement. MAWRs were treated with a standard or MOI-containing diet. Young rats (YWR, 16 weeks old) were the controls and received a standard diet. The hearts and aortas were harvested to evaluate SIRT1 and FOXO1 expression via Western blot and/or immunostaining, SIRT1 activity via a fluorometric assay, and oxidative stress using the DHE fluorescent probe. In the hearts and aortas, SIRT1 expression, reduced in MAWRs compared to YWRs, was enhanced in MOI MAWRs. In the hearts, SIRT1 activity did not differ between YWRs and MAWRs, whereas it was increased in MOI MAWRs compared with them. In the aortas, SIRT1 activity decreased in MAWRs, and it was similar in the MOI MAWRs and YWRs. FOXO1 expression increased in the nuclei of MAWR aortas compared to YWR and was reversed in MOI MAWRs. Interestingly, MOI treatment normalized oxidative stress enhanced in MAWRs, in both the heart and aorta. These results demonstrate the protective role of MOI against cardiovascular dysfunction due to aging via enhanced SIRT1 function and subsequently reduced oxidative stress.

## 1. Introduction

Over the years, aging research has outlined mechanisms and pathways that influence longevity and a healthy lifespan. Studies, first conducted in short-lived models and then in more complex organisms, have unveiled the role of (NAD+)-dependent deacetylases, called Sirtuins and Forkhead box “O” (FOXO) transcription factors, in modulating cellular functions deregulated during aging [1].

Sirtuin 1 (SIRT1), the best-studied member of Sirtuins, acts as a cellular stress sensor by activating adaptive responses under stressful conditions, such as nutrient deprivation and hypoxia, as an increased oxidant and pro-inflammatory species [2].

The lost cardiovascular system homeostasis that occurs inexorably during aging is particularly detrimental to the organism. Blood vessels become more vulnerable to the stressogenic stimuli that, if not adequately counteracted, can promote cellular dysfunction and cardiovascular disease development even without risk factors [3].

Sirtuins and FOXOs, of which the levels are reduced in aged blood vessels, play a coordinated role in maintaining vasculature homeostasis [4]. In fact, post-translational modifications, such as acetylation, modulate FOXO function and make cells capable of reacting to environmental stimuli [5].

Among the four isoforms of FOXOs identified in mammals thus far, FOXO1, FOXO3, and FOXO4 are ubiquitous with dynamic and variable expression levels, depending on the tissue and developmental stage. FOXO1 is expressed at a high level in the blood vessels and plays a crucial role in modulating vessel development [6]. In contrast to FOXO3 or FOXO4 mice, embryonic FOXO1-knockout mice show an impaired vasculogenesis, leading to death [7]. Notably, an important role of FOXO1 has also been demonstrated in angiogenesis and postnatal neovascularization [8].

SIRT1 and SIRT1/FOXO1 pathways also mediate the antioxidant effect of bioactive extracts containing phenolic and other natural compounds [9,10].

Most evidence has been accumulated regarding resveratrol, mimicking the transcriptional effect of caloric restriction and physical exercise, inducing SIRT1 activity and the subsequent anti-aging action [11,12,13]. Resveratrol can mitigate aging in various tissues, including the aorta and heart, where it has been reported to attenuate senescence-related apoptotic signaling and oxidative stress by modulating the SIRT1/FOXO1 pathway [14].

Other plants containing compounds with antioxidant and anti-inflammatory vascular effects are currently used in Asian and African traditional medicine. One of them is *Moringa oleifera* (MOI), called the ben nut tree, of which the leaves, seeds, roofs, and flowers are used by the Malagasy population as food and a remedy to overcome numerous diseases. In Malagasy traditional medicine, the leaves and flowers of this plant are used to relieve a cough and hypertension, especially in the northern part of Madagascar [15].

Recently, it has been shown that treatment with MOI leaf extract can improve the redox state in C2C12 skeletal muscle cells by increasing the activity of antioxidant enzymes, the known targets of SIRT1, such as manganese superoxide dismutase (SOD2) and catalase [16].

We previously described in spontaneously hypertensive rats (SHRs) treated with MOI-seed-powder-attenuated cardiac and vascular impairments associated with high blood pressure, reducing nocturnal heart rate, left ventricle remodeling, and heart fibrosis through a mechanism involving peroxisome-proliferator-activated (PPAR)α receptor expression [17]. MOI was able to reduce both oxidative stress and inflammation by decreasing the expression of proteins such as NADPH oxidase 2 subunits (p22^phox^ and p47^phox^), inducible nitric oxide synthase (iNOS), and nuclear factor kappa-light-chain-enhancer of activated B cells (NF-κB), and by upregulating the SOD2 in SHR aortas. This was associated with reduced free 8-isoprostane plasmatic levels [18]. Moreover, in both the aorta and the mesenteric arteries of middle-aged Wistar rats (MAWRs), the endothelial function was enhanced by 4 weeks of MOI treatment due to the increased nitric oxide (NO)-dependent or endothelial-derived hyperpolarizing factor (EDHF)-dependent relaxation, respectively. This was associated with the increased endothelial nitric oxide synthase (eNOS) activation through Akt signaling, and the downregulation of arginase-1 expression [19].

Similar vascular beneficial effects were previously associated with the improved SIRT1 activity by other natural compounds [20,21,22].

Thus, the present study aimed to investigate the possible contribution of SIRT1 signaling in modulating the beneficial effects of MOI treatment on the heart and aorta of MAWRs.

## 2. Results

### 2.1. Phytochemical Analysis of the Ethanolic Extract from MOI Seeds

First, we aimed to identify the substances contained in MOI seeds that could mediate the previously observed beneficial cardiovascular effects [17,18,19]. For this purpose, we performed a qualitative analysis of the constituents of the total MOI ethanolic extract (Table 1).

This analysis revealed the presence of alkaloids, polysaccharides, saponins, steroids, and terpenoids, that could be implicated in the observed cardiovascular effects. In contrast, we did not detect flavonoids, polyphenols, and tannins in the total extract of the MOI seeds, indicating the absence or a non-detectable concentration of these substances in the seeds. Thus, the beneficial effects observed were not due to these phytochemical constituents.

Furthermore, our analysis did not detect the presence of polyphenols, i.e., the known activators of SIRT1. Nevertheless, other substances in the total ethanolic extract were reported to be equally capable of modulating SIRT1 function. In particular, ginsenosides, a class of triterpene saponins with a steroidal structure, have been found to protect against multiple pathological conditions and senescence by regulating SIRT1 signaling [23].

The effects of the MOI extract on the endothelial NO release in YWR aortas were evidenced (Appendix A).

### 2.2. MOI Treatment Induces SIRT1 Expression and Activation in MAWR Cardiovascular Tissues

We evaluated the effect of MOI administration on SIRT1 expression and activity in the heart and aorta of the rats.

In the heart, SIRT1 expression decreased in MAWRs compared to YWRs, and slightly increased in the MOI MAWRs, without reaching statistical significance. Notably, there was no statistically significant difference between YWRs and MOI MAWRs (Figure 1a). SIRT1 activity did not differ between YWRs and MAWRs, whereas it increased considerably in the MOI MAWRs compared to YWRs and MAWRs (Figure 1b).

In the aorta, both the expression (Figure 2a) and activity (Figure 2b) of SIRT1 decreased in MAWRs compared to YWRs, and increased dramatically in MOI MAWRs compared to MAWRs, reaching levels similar to those measured in YWRs. No statistically significant difference between the MOI MAWRs and YWRs was found.

To confirm the results obtained in the aorta via Western blot analysis and fluorometric assay, we investigated the expression and nuclear migration of SIRT1 in the aortic rings through immunofluorescence labeling.

In the YWR aortic rings, SIRT1 was expressed and activated in the nuclei of aortic cells, particularly in the media layer and the endothelium, while in the MAWR aortas, the expression and the presence of the SIRT1 in the nuclei were strongly reduced (Figure 3a,a’) as an effect of aging. The treatment of MOI for 4 weeks in MAWRs was able to restore the expression and presence in the aortic nuclei of SIRT1 to levels similar to the YWR aortas (Figure 3a,a’), reverting the age-associated impairment.

We also evaluated the immunofluorescence of SIRT1 in the cell nuclei (white spots) with the Fiji-win32 software in confocal images of aortas, confirming reduced SIRT1 levels in MAWR aortas and a significant increase in SIRT1 nuclear presence in MOI MAWRs (Figure 3b).

### 2.3. MOI Treatment Reduces FOXO1 Expression and Activation in the Aorta of MAWRs

Because SIRT1 deacetylates FOXO1, modulating the activation of this transcription factor, we evaluated FOXO1 expression and activation through immunofluorescence labeling of aortic sections.

In YWR aortic rings, FOXO1 was expressed in the cytoplasm, while in MAWR aortas nuclear FOXO1 expression was enhanced (Figure 4a,a’). The treatment of MOI for 4 weeks in MAWRs was able to reduce nuclear FOXO1 expression in aortas to levels comparable to YWR aortas (Figure 4a,a’).

We also evaluated the immunofluorescence of FOXO1 in the cell nuclei in confocal images of aortas, confirming the reduction in FOXO1 expression in both YWR and MOI MAWR aortas, and a significant increase in FOXO1 expression in untreated MAWRs (Figure 4b).

### 2.4. MOI Treatment Reduces Oxidative Stress in MAWRs

To test whether the increased activity of SIRT-1 evidenced in the hearts and aortas of MOI MAWRs was associated with a reduction in oxidative stress, we directly assessed the in-situ production and topographical distribution of superoxide anion (O_2_^−^) in both the heart and aortic sections from YWRs, MAWRs and MOI MAWRs. As expected, compared to YWRs, MAWRs showed a marked red fluorescence associated with EtBr formation from a DHE oxidized probe due to aging (Figure 5a,b, respectively). In contrast, ROS staining in both the heart and aorta of MOI MAWRs was comparable to the YWR organs (Figure 5a,b, respectively).

## 3. Discussion

Our previous study described the beneficial effects of MOI seed administration in both the aorta and mesenteric arteries of MAWRs with an established aging-induced endothelial dysfunction. MOI treatment in MAWRs was able to boost eNOS activity in the aorta and EDHF-dependent relaxation in mesenteric arteries [19]. In the present study, we first performed a phytochemical analysis of MOI seeds, identifying alkaloids, polysaccharides, steroids, terpenoids and saponins as the components of the total ethanolic extract.

Although our analysis did not reveal the presence of polyphenols, the known activators of SIRT1, some of the compounds present in MOI seeds have been suggested as potential modulators of SIRT1 function. In MAWRs, we previously reported reduced endothelium-dependent NO relaxation due to altered Akt/eNOS signaling [19]. MOI administration corrected the Akt/eNOS defect and restored NO-mediated, endothelium-dependent relaxation in MAWR aortas [19]. It has been reported that saponins, triterpenes and polysaccharides improve endothelial function by interfering with calcium handling, PI3K/Akt/eNOS and angiotensin II signaling in rodents [24,25,26].

Moreover, according to Yang and colleagues, saponins contained in *Gynostemma pentaphyllum* increase NO and antioxidant defense in mice subjected to choline-induced vascular dysfunction and oxidative stress [27]. Another study demonstrated the involvement of *Panax notoginseng* saponins in the inhibition of endothelial cell apoptosis induced by advanced glycation end-products. The suggested mechanism was the SIRT1 upregulation [28].

Alongside that, ginsenosides, a class of triterpene saponins with a steroidal structure, have been found to protect against multiple pathological conditions and senescence by upregulating SIRT1 signaling [23].

Based on these data, we decided to evaluate whether the beneficial vascular effect of MOI seed treatment could be explained at least in part by the involvement of SIRT1 signaling.

We found that a 4-week treatment with MOI seeds was able to recover the SIRT1 expression and activity in MOI MAWRs, decreased in aged cardiovascular tissues, to levels comparable to those measured in young individuals in both the heart and aorta. This could contribute to vascular function improvement during aging, previously described in MOI MAWRs [19].

In addition, the ability of MOI seeds to induce NO release in rat aortic rings shown in the present work (Appendix A) also suggests the involvement of SIRT1, which is known to stimulate NO production through eNOS deacetylation [29,30].

Through immunofluorescence labeling, we demonstrated that in YWRs, SIRT1 was expressed in the aortic nuclei, particularly in the media layer and endothelium. The increased SIRT1 activity in the heart and aorta of MOI MAWRs was confirmed via a specific fluorometric assay. Inversely, FOXO1 was mainly expressed in the aortic cytoplasm of YWRs, while its expression was increased in the aortic nuclei in MAWRs. Of note, MOI treatment in MAWRs was able to reduce FOXO1 expression in aortic cell nuclei to levels comparable to those measured in YWRs.

It has been reported that SIRT1 modulates cell proliferation, survival and senescence, while also establishing a complex interaction with FOXOs, thus affecting clinical conditions such as cancer [31,32]. Notably, SIRT1-dependent deacetylation can influence the stability, subcellular localization and transcriptional activity of FOXOs, playing a crucial role in modulating vascular homeostasis [33,34].

When measuring the in situ production of O_2_^−^, we found that ROS levels, higher in MAWRs than YWRs, decreased in MOI MAWRs to levels comparable to YWRs, in both the heart and aortic sections. This trend in ROS levels could explain the much higher levels of FOXO1 in MAWRs than in YWRs and MOI MAWRs. In fact, it has been suggested that in both *C. elegans* and mammalian cells, an increased amount of ROS represents a stimulus for DAF-16/FOXO to translocate from the cytosol to the nucleus, acting as a negative feedback control for ROS, in order to counteract the accumulation of oxidative stress and prevent cellular damage [35].

Moreover, it has been proposed that the ability of SIRT1 to modulate FOXO transcriptional activity, aimed at counteracting oxidative stress, could depend on the amount of ROS and the duration of cellular exposition to such a stressogenic stimulus. For instance, in *C. elegans* it has been demonstrated that, in the presence of low oxidative stress, FOXOs are activated via phosphorylation, while at higher concentrations or longer periods, they can be inactivated via acetylation. The deacetylation operated by Sir2 (the ortholog of *SIRT1* in *C*. *elegans*) has been shown to prolong the FOXO activity induced by treatment with H_2_O_2_, in order to ensure the full activation of the antioxidant targets of FOXO.

A limitation of the present study is that we did not measure the levels of acetylated FOXO1, and consequently, we could not demonstrate which form of FOXO1 (acetylated or not) is expressed at higher levels in the nuclei of the aorta of MAWRs compared to that in MOI MAWRs. Further studies are needed to clarify this point, to determine whether SIRT1 is directly involved in the shuttling of FOXO1 between the cytoplasm and nucleus, and the link with the observed antioxidant effects.

The phytochemical analysis of the MOI seeds failed to detect flavonoids, polyphenols and tannins, while revealing the presence of alkaloids, polysaccharides, saponins, steroids and terpenoids, that could be implicated in the observed cardiovascular effects. The observed beneficial cardiovascular effects could be due to a synergistic action of several molecules contained in MOI seeds. Based on available literature data, we suggested that the effects of MOI, associated with SIRT1 expression and activity, could be due to triterpenes and saponins, which have indeed been reported to enhance SIRT1 signaling [23,28].

We tested whether the MOI alkaloid fraction alone was able to induce the potential release of vasorelaxant agents in conductance arteries such as the aorta by measuring NO release in the aortic rings using Fe(DETC)2 as a spin trap and EPR. The vessels were able to release NO when incubated with a spin trap solution containing 100 mg of the total ethanolic extract of MOI (Appendix A). In contrast, the alkaloid fraction failed to induce NO release (data not shown). We have not yet tested the effect of the saponin fraction alone on NO induction; therefore, further investigation is needed to verify this potential effect.

In conclusion, our results demonstrate that MOI treatment is able to increase the expression and activity of SIRT1 in both the heart and aorta of MAWRs, and that this is associated with a significant decrease in the amount of ROS accumulated in cardiovascular tissues during aging, partially explaining the previously observed beneficial cardiovascular effects after MOI treatment [17,18,19].

## 4. Materials and Methods

### 4.1. Phytochemical Screening of the MOI Seed Ethanolic Total Extract

Extraction preparation: Fresh MOI pods were peeled, and 100 g of seeds were ground using a porcelain mortar with a pestle. The seed powder was macerated with 96% ethanol two times within 24 h, and then filtrated with absorbent cotton. The filtrate was concentrated in a vacuum rotary evaporator (Heidolph, Germany) and the bath temperature was regulated at 40 °C. At the end of evaporation, 7.3 g of extract was obtained and stored at −20 °C until use.

In order to reveal the constituents of the MOI extract, the methods used by Fong and co-workers, Aiyegoro and Okoh, were adopted with a few modifications [36,37]. After dissolving the extract in chlorhydric acid (HCl) 1%, the alkaloids precipitated with the Drangendorff reagent (VWR, France). To detect flavonoids, the extract was first dissolved in methanol and heated. The appearance of red or orange coloration after the addition of magnesium metal turnings followed by a few drops of concentrated HCl indicated the presence of flavonoids. The saponins were revealed by the presence of emulsion after dissolving the extract in distilled water and vigorous shaking.

For terpenoids and steroids, the extract was dissolved in chloroform. After filtration, concentrated sulfuric acid was added and the occurrence of a reddish-brown interface coloration characterized these products. A filtrate solution of the extract in distilled water was made and a ferric chloride reagent was added. Blue coloration indicated the presence of tannins. The extract was dissolved in distilled water, heated and agitated. Four drops of sodium chloride 10% were added, then 4 to 5 drops of gelatin 1%. Precipitation indicated the presence of polyphenols. A decoction was prepared with MOI seed powder and then filtered. Three volumes of alcohol were added, and the polysaccharides were marked via precipitation. The sign (+) was applied to characterize the presence and (-) for the absence of the phytochemical constituents.

### 4.2. Animal Model

Male Wistar young rats (YWRs, 16 weeks old) were treated for 4 weeks with a standard diet, corresponding to the study’s control group. Middle-aged Wistar rats (46 weeks old) were treated for 4 weeks with a standard diet (MAWRs) or a diet containing the powder of MOI seeds (750 mg/kg/day, MOI MAWRs). This dose was used in our previous studies and is currently used in rodent models [18,19]. All the groups of rats received water ad libitum. The rats were then sacrificed (YWRs at 20 weeks of age, MAWRs and MOI MAWRs at 50 weeks of age) to harvest the hearts and aortas for Western blot, fluorometric analysis and immunostaining. All the experiments were conducted as per the agreement of the Ethical Committee “Comitato Etico Campania Sud” (Prot.n.4_r.p.s.o.) and the Ethical Committee Guide for Care and Use of Laboratory Animals of Nantes University (authorization number 00909.01).

### 4.3. Western Blot Analysis

Total proteins were extracted from the aortic and heart tissues using a RIPA buffer with protease inhibitors, and quantified using the Bradford protein assay.

A total of 45 μg protein per well was separated via electrophoresis on 4–15% sodium dodecyl sulfate polyacrylamide gels and transferred to nitrocellulose membranes. The membranes were then blocked with 5% milk TBST buffer (TBS plus 0.1% Tween-20) for 1 h at room temperature, and incubated with the primary antibody, anti-SIRT1 (D1D7) (Cell Signaling Technology, Inc., Danvers, MA, USA) or anti-GAPDH (G-9) (Santa Cruz Biotechnology, Inc., Dallas, TX, USA) overnight at 4 °C, washed three times with the TBST buffer and incubated with the corresponding secondary antibodies at room temperature for 45 min, anti-rabbit and anti-mouse IgG heavy and light chain (Bethyl Laboratories, Inc., Montgomery, TX, USA), respectively. The signals were detected using the “Enhanced Chemiluminescent Substrate” method with ECL Star (EuroClone S.p.a, Milan, Italy) and analyzed with the ImageLab software, using the Chemidoc image acquisition and analysis tool (Bio-Rad Laboratories, Inc., Hercules, CA, USA). All data were expressed as the mean ± SD.

### 4.4. Nucleus Extraction and SIRT1 Activity Assay

The nuclei were extracted from the heart tissue using a commercial kit (EpiGentek Group Inc., New York, NY, USA), and from aortic pooled samples using a lysis buffer containing 50 mM Tris-HCl (pH 7.4), 150 mM NaCl, 0.5% Na-deoxycholate and 100 μM Na-orthovanadate. After incubation on ice for 30 min, the heart and aortic lysates were centrifugated at 13,000 rpm for 20 min at 4 °C. The protein extracts were quantified using the Bradford protein assay.

SIRT1 activity was measured using a deacetylase fluorometric assay kit (Sir2 Assay Kit, CycLex, Ina, Nagano, Japan), following the manufacturer’s instructions. The values were reported as relative fluorescence/µg to proteins (AU). All data were expressed as the mean ± SD.

### 4.5. Staining and Confocal Microscopy Imaging

Frozen sections of aortas (7 µm thick) were fixed on glass slides with cold 100% methanol and incubated (2 h at room temperature) in a blocking buffer (5% non-fat dry milk in PBS). The aortic sections were then incubated overnight (4 °C), with either a monoclonal murine anti-SIRT1 or anti-FOXO1 (1:100 and 1:50, Elabscience, Houston, TX, USA) for SIRT1 or FOXO1 immunostaining, respectively. Three washes were followed by incubation (1 h at room temperature in the dark) with the secondary rabbit fluorescent Alexa fluor-647-conjugated antibody (1:500; Thermo Fisher Scientific, Illkirch, France). The nuclei were stained blue using DAPI. A Nikon A1-RS inverted laser scanning confocal microscope (Nikon Instruments Europe BV, Amsterdam, The Netherlands) was used for the optical sectioning of the tissue. Digital image recording was performed using the NIS element software (Nikon Instruments Europe BV, Amsterdam, The Netherlands). The images were analyzed and processed using the Fiji software (https://imagej.net/, accessed on 22 August 2022). The software was used to calculate the intensity and the number of white spots in the nuclei, corresponding to the overlapping of DAPI fluorescence and SIRT1 or FOXO1 fluorescence. The signal was normalized to the same number of the cell nuclei analyzed, and chosen randomly in different parts of the vascular wall. In another set of experiments, frozen sections of the heart and aorta (7 µm thick) were used for the in situ detection of O_2_^−^, using the oxidative fluorescent probe, dihydroethidium (DHE, Sigma-Aldrish, Saint Quentin Fallavier, France), that shows red fluorescence when oxidized to EtBr [38].

### 4.6. Data Analysis

The results were expressed as the mean ± SEM of n, where n represents the number of rats used for each experiment. The SIRT1 or FOXO1 expression levels and/or activity were compared using a one-way analysis of variance, followed by a Bonferroni multiple comparison post hoc test, after a Ficher test, to confirm the equal variance for the statistical parametric test application. Statistical significance was set at * *p* < 0.05, ** *p* < 0.01 and *** *p* < 0.001. The graphical representation of the data and the statistical analyses was carried out using Stata version 16.0 (Stata Corporation, College Station, TX, USA).

## Figures and Tables

**Figure 1 pharmaceuticals-16-00761-f001:**
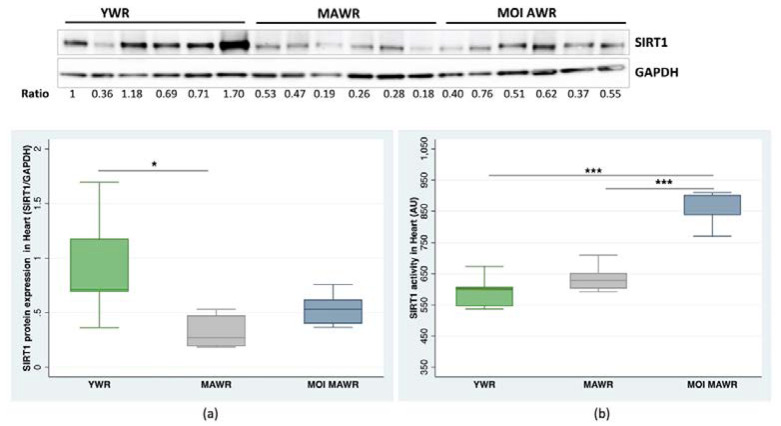
Sirtuin 1 (SIRT1) protein expression and activity were measured in the heart of young rats (YWRs, 16 weeks old), control middle-aged rats (MAWRs, 50 weeks old), and MAWRs treated with *Moringa oleifera* seeds (MOI MAWRs). (**a**) The representative Western blot and corresponding densitometric analysis of SIRT1 expression normalized to GAPDH expression in the heart of YWRs, MAWRs, and MOI MAWRs (* *p* < 0.05, *n* = 6 for each group). (**b**) SIRT1 activity in the heart of YWRs, MAWRs, and MOI MAWRs (*** *p* < 0.001, *n* = 6 for each group).

**Figure 2 pharmaceuticals-16-00761-f002:**
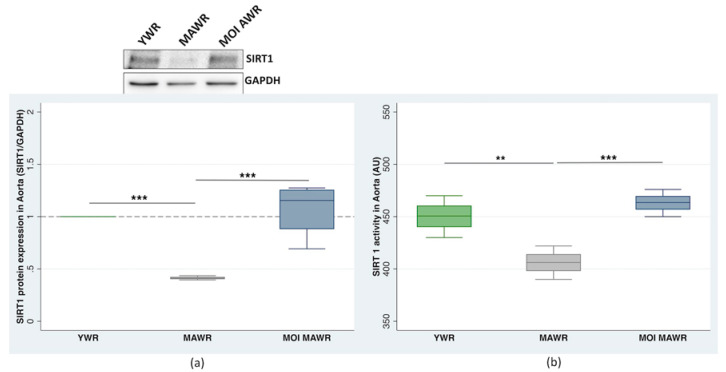
Sirtuin 1 (SIRT1) protein expression and activity were measured in the aorta of young rats (YWRs, 16 weeks old) and control middle-aged rats (MAWRs, 50 weeks old) and MAWRs treated with *Moringa oleifera* seeds (MOI MAWRs). (**a**) The representative Western blot and corresponding densitometric analysis of SIRT1 expression normalized to GADPH in the aorta of YWRs, MAWRs, and MOI MAWRs (*** *p* < 0.001), pooled cell lysate from five rats for each group. (**b**) SIRT1 activity in the aorta of YWRs, MAWRs, and MOI MAWRs (** *p* = 0.01; *** *p* = 0.001), from five rats of each group.

**Figure 3 pharmaceuticals-16-00761-f003:**
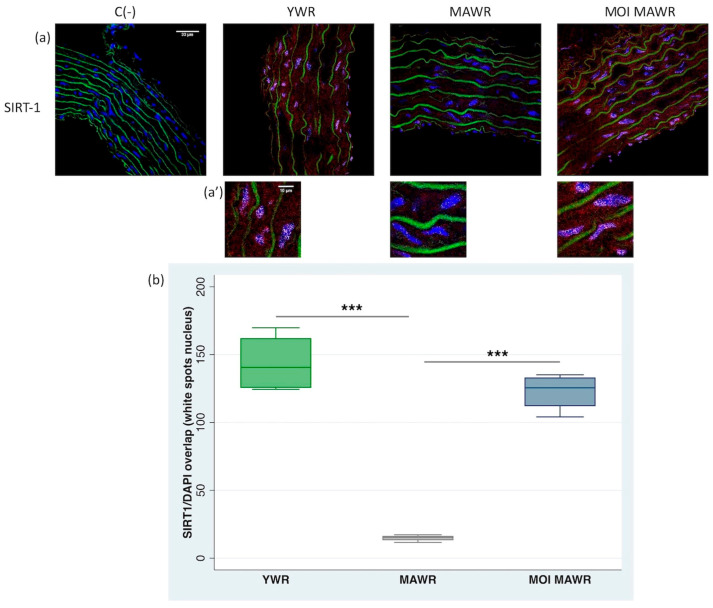
(**a**) Immunohistochemical analysis of SIRT1 in the aortas from young Wistar rats (YWRs, 20 weeks old) or middle-aged rats (MAWRs, 50 weeks old) after 4 weeks of normal diet or a diet containing Moringa oleifera seed powder (MOI MAWRs). The red fluorescence in the cells is relative to the cytoplasmic localization of SIRT1. The green fluorescence corresponds to elastin autofluorescence. C- negative control without primary antibody incubation. Bar = 33 µm. (**a’**) The white spots seen in the cell nuclei correspond to the overlapping of DAPI fluorescence and SIRT1 immunostaining. Bar = 10 µm. (**b**) The number and intensity of the white spots in the cell nuclei corresponding to the DAPI/SIRT1 overlap were evaluated with the Fiji-win32 software using aorta images and normalized to the same elevated number of nuclei chosen randomly and in all the vascular walls. *n* = 4 for each group, *** *p* < 0.01 MAWRs versus YWRs or MOI MAWRs.

**Figure 4 pharmaceuticals-16-00761-f004:**
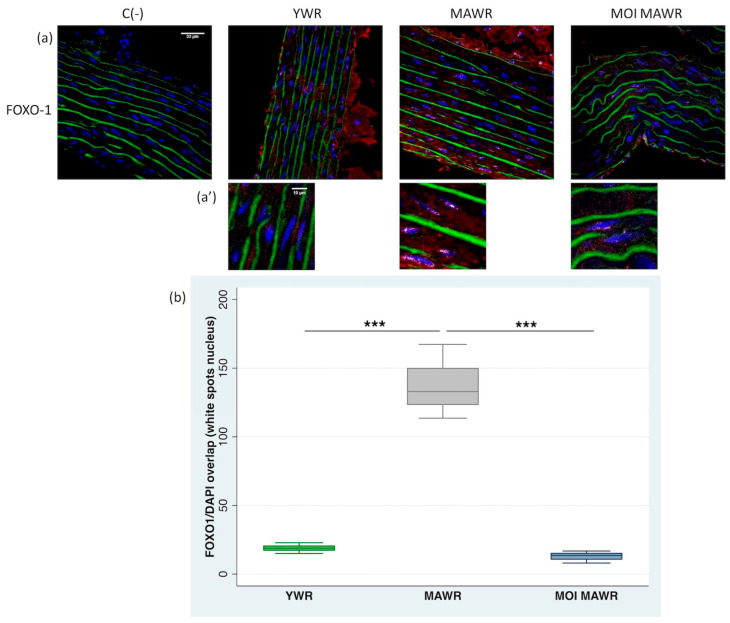
(**a**) Immunohistochemical analysis of FOXO1 in aortas from young Wistar rats (20-week-old rats, YWRs) or middle-aged rats (MAWRs, 50 weeks old) after 4 weeks of normal diet or a diet containing Moringa oleifera seed powder (MOI MAWRs). The red fluorescence in the cells is relative to the cytoplasmic localization of FOXO1. The green fluorescence corresponds to elastin autofluorescence. C- negative control without primary antibody incubation. Bar = 33 µm. (**a’**) The white spots seen in the cell nuclei correspond to the overlapping of the DAPI fluorescence and FOXO1 immunostaining. Bar = 10 µm. (**b**) The number and intensity of the white spots in the cell nuclei corresponding to the DAPI/FOXO1 overlap were evaluated with the Fiji-win32 software using aorta images and normalized to the same number of the nuclei chosen randomly and in all the vascular walls. *n* = 4 for each group, *** *p* < 0.001 MAWRs versus YWRs or MOI MAWRs.

**Figure 5 pharmaceuticals-16-00761-f005:**
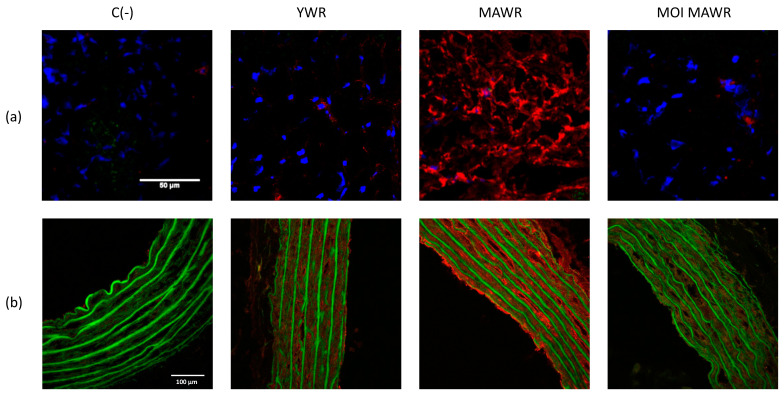
Qualitative oxidative stress evaluation (red staining by DHE) of O_2_^−^ in the hearts (**a**) and aortas (**b**) of young Wistar rats (YWRs, 20 weeks old) or middle-aged rats (MAWRs, 50 weeks old) after 4 weeks of normal diet or a diet containing Moringa oleifera seed powder (MOI MAWRs). Blue nuclear staining of the hearts was obtained with DAPI, bar = 50 µm. In the aortas, the green fluorescence corresponds to elastin autofluorescence, bar = 100 µm. Negative control C (-) without DHE, *n* = 3 for each group.

**Table 1 pharmaceuticals-16-00761-t001:** Phytochemical constituents of the MOI seed extract.

Phytochemical Constituents	Results
Alkaloids	+
Flavonoids	-
Polysaccharides	+
Polyphenols	-
Saponins	+
Steroids	+
Tannins	-
Terpenoids	+

## Data Availability

Data is contained within the article and Appendix A.

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
