# Peer review of "SIRT1 Signaling Is Involved in the Vascular Improvement Induced by Moringa Oleifera Seeds during Aging"

_pharmaceuticals, 2023, doi:10.3390/ph16050761_

Round 1

Reviewer 1 Report

Dear Authors,

please address the following issues.

Figure 1: the immunoblottings described in figure 1a must be better presented. A legend should show the significance of each lane. There are 6 lanes but, in the figure legend, it is stated that there are 5 mice each group. Please clarify. The fact that the enzymatic activity of SIRT1 increases also in the MAWR mice even in absence of MOI should be commented. The figure as it is does not support the results and should be reevaluated after the legend has been inserted.

Figure 2: the results obtained on the aorta cells are consistent.

Discussion: lines 264-266. The modulation of FOXOs by SIRT1 is known and described mostly in cancer. SIRT1 is also a downstream target of hypoxia inducible factor 1alfa (HIF1alfa) and is activated by c-Myc by promoting NAMPT transctiption and NAD-salvage pathway, for instance. These  are examples of pivotal pathways and interactions that affect cell proliferation, survival, drug resistance (Menssen A. et al. PNAS 2012; Frazzi R Frontiers Endocrinol 2018). These aspects must be cited and commented.

General issues: the format of the document says "International Journal of Molecular Sciences" whereas the review is for the Journal "Pharmaceuticals". Please change it.

Minor issues: please, check the grammar and punctuation throughout the manuscript. For example, the space between"Figure" and "5" (line 214) and so on. 

The English is fine. Minor grammar and punctuation changes are required.

Author Response

please address the following issues.

Figure 1: the immunoblottings described in figure 1a must be better presented. A legend should show the significance of each lane. There are 6 lanes but, in the figure legend, it is stated that there are 5 mice each group. Please clarify.

Reply: We would thank the reviewer for the comment. Following the suggestion, we modified Figure 1 accordingly, introducing the SIRT1/GAPDH ratio values for each lane. We also beg pardon for the mistake in the number of mice, we also corrected it.

The fact that the enzymatic activity of SIRT1 increases also in the MAWR mice even in absence of MOI should be commented. The figure as it is does not support the results and should be reevaluated after the legend has been inserted. 

Reply: Indeed, SIRT1 activity mean values of YWR and MAWR are almost the same (593.15±55.05 vs 635.5±42.84, respectively) as shown in Figure 1. To better illustrate this, we reinforced the line of the mean value in YWR.

Discussion: lines 264-266. The modulation of FOXOs by SIRT1 is known and described mostly in cancer. SIRT1 is also a downstream target of hypoxia inducible factor 1alfa (HIF1alfa) and is activated by c-Myc by promoting NAMPT transcription and NAD-salvage pathway, for instance. These are examples of pivotal pathways and interactions that affect cell proliferation, survival, drug resistance (Menssen A. et al. PNAS 2012; Frazzi R Frontiers Endocrinol 2018). These aspects must be cited and commented. 

Reply: We thank the Reviewer for her/his helpful suggestion. We added the references to stress the involvement of SIRT1/FOXO pathway in modulating important cellular functions and clinical conditions. In the discussion, the following sentence was included: “It has been reported that SIRT1 modulates cell proliferation, survival, and senescence, while also establishing a complex interaction with FOXOs, thus affecting clinical conditions such as cancer [31,32].

General issues: the format of the document says "International Journal of Molecular Sciences" whereas the review is for the Journal "Pharmaceuticals". Please change it. 

Reply: We beg pardon, we corrected it.

Minor issues: please, check the grammar and punctuation throughout the manuscript. For example, the space between"Figure" and "5" (line 214) and so on. The English is fine. Minor grammar and punctuation changes are required.  

Reply: OK

Reviewer 2 Report

In this paper, the authors investigated the involvement of SIRT1 in Moringa oleifera (MOI) -induced vascular improvement. Their results suggest a protective role of MOI against cardiovascular dysfunction due to aging via enhanced SIRT1 function and subsequent reduced oxidative stress. Here are my questions/ comments.

1.       Results 2.1, the authors mentioned that other substances besides polyphenols in the total ethanolic extract were reported to be equally capable to modulate SIRT1 function. Please add the related references in the manuscript.

2.       As the authors demonstrated that there were multiple constituents in MOI seed extract, which did the authors believe exerted the main function? Did the authors conduct experiments with a pure single constituent with or without its corresponding antagonist?

3.       Based on Figure 1 and Figure 2, SIRT1 expression in hearts and aortas, was reduced in middle-aged Wistar rats (MAWR) compared to young rats (YWR). The SIRT1 expression in the aortas but not in the hearts was enhanced in MOI MAWR. Please modify the abstract stating that in hearts and aortas SIRT1 expression, which was reduced in MAWR compared to YWR, was enhanced in MOI MAWR.

4.       Throughout the paper, the authors investigated how MOI induced vascular improvement in the MAWR, as the controls, how about the effect of MOI in the YWR?

Author Response

  1. Results 2.1, the authors mentioned that other substances besides polyphenols in the total ethanolic extract were reported to be equally capable to modulate SIRT1 function. Please add the related references in the manuscript.

We thank the Reviewer for her/his comments and suggestions. In the results 2.1 we suggested the class of triterpene saponins as molecules potentially involved in the modulation of SIRT1 signaling (Lou et al. 2021, reference 23 cited in the manuscript). Furthermore, in the discussion section of the manuscript, we cited the work published by Bo et al. in 2020 (reference 28 in the manuscript) suggesting saponins as molecules implicated in the anti-oxidant and anti-apoptotic endothelial effects of Panax notoginseng involving a SIRT1 mechanism.

  1. As the authors demonstrated that there were multiple constituents in MOI seed extract, which did the authors believe exerted the main function? Did the authors conduct experiments with a pure single constituent with or without its corresponding antagonist?

We agree with the Reviewer on the importance to stress this issue. Accordingly, we added the following paragraph in the discussion section: “The phytochemical analysis of MOI seeds failed to detect flavonoids, polyphenols, and tannins, while revealing the presence of alkaloids, polysaccharides, saponins, steroids, and terpenoids that could be implicated in the observed cardiovascular effects. The observed beneficial cardiovascular effects could be due to a synergistic action of several molecules contained in MOI seeds. Based on available literature data, we suggested that the effects of MOI, that are associated with SIRT1 expression and activity, could be due to the triterpene saponins, which indeed have been reported to enhance SIRT1 signaling.

We tested whether the MOI alkaloid fraction alone was able to induce the potential release of vasorelaxant agents in conductance arteries such as the aorta by measuring NO release in the aortic rings using Fe(DETC)2 as a spin trap and EPR. The vessels were able to release NO when incubated with a spin trap solution containing 100 mg of the total ethanolic extract of MOI (supplementary data). In contrast, the alkaloid fraction failed to induce NO release (data not shown). We have not yet tested the effect on NO induction of the saponin fraction alone, therefore further investigation is needed to verify this potential effect”.

  1. Based on Figure 1 and Figure 2, SIRT1 expression in hearts and aortas, was reduced in middle-aged Wistar rats (MAWR) compared to young rats (YWR). The SIRT1 expression in the aortas but not in the hearts was enhanced in MOI MAWR. Please modify the abstract stating that in hearts and aortas SIRT1 expression, which was reduced in MAWR compared to YWR, was enhanced in

We thank the Reviewer and modified the abstract according to her/his advice.

  1. Throughout the paper, the authors investigated how MOI induced vascular improvement in the MAWR, as the controls, how about the effect of MOI in the YWR?

We thank the Reviewer for her/his comment. Although evaluating the potential effects of MOI on young individuals represents an interesting issue, we used MAWR as the control because the aim of the study was to investigate how MOI induced improvement of aged-associated vascular dysfunction.

However, we previously tested natural substances on animal models without pathological state and we did not find any effect to improve cardiovascular function (Agouni A et al. Red Wine Polyphenols Prevent Metabolic and Cardiovascular Alterations Associated with Obesity in Zucker Fatty Rats (Fa/Fa). PLOS ONE 2009;4(5):e5557). We believe that to demask anti-inflammatory and anti-oxidant effects in cardiovascular tissues by agents contained in MOI it is necessary to have a pathological state such as age-associated vascular impairments. Undoubtedly, this issue could be deeply investigated but it was beyond the purpose of our study.

Round 2

Reviewer 2 Report

The authors have addressed my concerns properly. I have no further questions.